

# Parsimonious statistical learning models for low flow estimation

Johannes Laimighofer[1], Michael Melcher[2], and Gregor Laaha[1]

[1]Institute of Statistics, University of Natural Resources and Life Sciences (BOKU), Vienna, Austria
[2]Institute of Information Management, FH JOANNEUM – University of Applied Sciences, Graz, Austria

**Correspondence:** Johannes Laimighofer (johannes.laimighofer@boku.ac.at)

**Abstract.** Statistical learning methods offer a promising approach for low flow regionalization. We examine seven statistical learning models (lasso, linear and non-linear model based boosting, sparse partial least squares, principal component regression, random forest, and support vector machine regression) for the prediction of winter and summer low flow based on a hydrological diverse dataset of 260 catchments in Austria. In order to produce sparse models we adapt the recursive feature elimination for variable preselection and propose to use three different variable ranking methods (conditional forest, lasso and linear model based boosting) for each of the prediction models. Results are evaluated for the low flow characteristic $Q95$ ($Pr(Q > Q95) = 0.95$) standardized by catchment area using a repeated nested cross validation scheme. We found a generally high prediction accuracy for winter ($R^2_{CV}$ of 0.66 to 0.7) and summer ($R^2_{CV}$ of 0.83 to 0.86). The models perform similar or slightly better than a Top-kriging model that constitutes the current benchmark for the study area. The best performing models are support vector machine regression (winter) and non-linear model based boosting (summer), but linear models exhibit similar prediction accuracy. The use of variable preselection can significantly reduce the complexity of all models with only a small loss of performance. The so obtained learning models are more parsimonious, thus easier to interpret and more robust when predicting at ungauged sites. A direct comparison of linear and non-linear models reveals that non-linear relationships can be sufficiently captured by linear learning models, so there is no need to use more complex models or to add non-liner effects. When performing low flow regionalization in a seasonal climate, the temporal stratification into summer and winter low flows was shown to increase the predictive performance of all learning models, offering an alternative to catchment grouping that is recommended otherwise.

## 1 Introduction

Estimating long term averages of low flow in ungauged basins are crucial for a wide range of applications, e.g. water resources management and engineering, hydropower planing or ecological issues (Smakhtin, 2001). The two main approaches for predicting low flow indices are either based on physical-based models (e.g. Euser et al., 2013), or statistical models. Statistical low flow models can be further subdivided into geostatistical models (e.g. Castiglioni et al., 2009, 2011; Laaha et al., 2014) and regression based methods (e.g. Laaha and Blöschl, 2006, 2007); an overview is given by Salinas et al. (2013). Regression methods cover a wide spectrum of models and especially in the last decade there was gaining interest in statistical learning models in hydrology (Abrahart et al., 2012; Dawson and Wilby, 2001; Nearing et al., 2021; Solomatine and Ostfeld, 2008). The applications include rainfall-runoff modeling by neural networks (e.g. Kratzert et al., 2019a, b), using Support Vector



Machines (SVM) for prediction of karst tracers (Mewes et al., 2020) or reference evapotranspiration (Tabari et al., 2012) and random forest for flood event classification (Oppel and Mewes, 2020). Nevertheless, the implementation of statistical learning methods for predicting low flow is still rare.

The considered methods so far can be classified into linear and non-linear statistical learners. Linear methods include, beside ordinary least squares regression approaches (OLS, Kroll and Song, 2013; Zhang et al., 2018; Ferreira et al., 2021), also linear models with a penalization parameter like elastic net (Worland et al., 2018), and linear boosting models (Tyralis et al., 2021). Further approaches are based on dimension reduction techniques such as partial least squares regression (PLS, Kroll and Song (2013)) or principal component regression (PCR, Kroll and Song, 2013; Nosrati et al., 2015). An example of non-linear ex-

tensions to the linear model is the Enhanced adaptive regression through hinges (Earth, Ferreira et al., 2021). Furthermore, various forms of tree-based methods were applied in low flow prediction, as Random Forest (RF, Ferreira et al., 2021; Zhang et al., 2018; Worland et al., 2018), gradient boosting with tree stumps (Tyralis et al., 2021) or M5-cubist (Worland et al., 2018). Additionally, the most comparative study of Worland et al. (2018), used an ensemble learning technique called meta M5-cubist, a non-linear kernel extension of K-nearest neighbour (KKNN) and two variants of support vector machines (polynomial and

gaussian kernel).

Given the large number of learning methods, it is a priori unclear which method will perform best for a particular study area. Only few studies have conducted a comparative assessment, typically focusing on single methods or a particular group of learners (Kroll and Song, 2013; Zhang et al., 2018; Worland et al., 2018; Ferreira et al., 2021). Comparing only linear models (PLS, PCR, OLS) Kroll and Song 2012 could not find any superior model for 130 stations in the eastern USA. Tree based

methods performed better in terms of point prediction for the CAMELS data set (Tyralis et al., 2021) or an Australian data set of 605 stations (Zhang et al., 2018). Ferreira et al. (2021) showed that Earth and RF perform similar for 51 stations in Brazil and Worland et al. (2018) achieved good performance in terms of a root mean squared error (RMSE) for the ensemble learning model analysing 224 stations in south-eastern USA. All of these studies were conducted for different hydroclimatic settings, but none did focus on a seasonal climate, where low flows in summer and winter are generated by different processes and

should be assessed separately. Such an assessment is missing and will be addressed in this study.

A general tendency visible from most studies is, that more complex models seem to perform better than more parsimonious ones, making model interpretation difficult and plausibility of parameters hard to judge. This leads to a major criticism of statistical learning models, as they are often inferred as "black box" models (Efron, 2020; See et al., 2007), meaning that prediction accuracy and process understanding cannot be reached at the same time (Kuhn and Johnson, 2019). One major approach of

improving the interpretability of statistical learning models, can be summarized under the concept of variable selection. Variable selection is a wide field of research, where we can basically identify three major approaches: (i) model inherent variable selection as in e.g. lasso, boosting models, or tree based methods; (ii) filter based methods (Guyon and Elisseeff, 2003), which include correlation based variable selection or univariate regression filters; and finally (iii) wrapper based methods (Kohavi and John, 1997) such as recursive feature elimination (RFE) (Guyon et al., 2002; Granitto et al., 2006) or genetic algorithms (Kuhn

and Johnson, 2019). Model inherent selection is a fast selection method, which main downside is that it is restricted to the underlying model. Filter based methods, where one advantage is usually the computation time, may suffer from weaker predictive





performance, as filtering options have no link to the final prediction model (Kuhn and Johnson, 2019; Guyon and Elisseeff, 2003). In contrast, wrapper methods have a higher computational burden, and greedy search algorithms as RFE may only find a local minimum, especially if interactions are present (Kuhn and Johnson, 2019). The RFE, like any variable selection method,

can suffer from a bias in the selection procedure (Ambroise and McLachlan, 2002), if poor validation strategies are chosen. Nevertheless, RFE can be an efficient technique to substantially reduce the predictor set (Kuhn and Johnson, 2019).

Although there appears to be a general consensus among hydrologists that parsimonious models offer a number of advantages over more complex models, including better parameter interpretability and robustness, surprisingly little effort has been undertaken to assess the merits of variable selection for statistical low flow regionalization. This is especially the case for sta-

tistical learning methods, which generally allow for higher complexity than regionalization approaches. Apart from stepwise regression procedures (e.g. Laaha and Blöschl, 2007; Kroll and Song, 2013), Tyralis et al. (2021) tested the inherent variable selection of the boosting algorithm and found the number of selected variables to be depending on the runoff characteristics to be predicted. Another approach, which uses at least inherent variable rankings for predictor variables, is to employ variable importance as the decrease in accuracy criterion of RF (Worland et al., 2018). They additionally used partial dependence plots,

which they found beneficial for analysing relationships between predictors and the response. However, these approaches can be misleading, when variables in boosting models are falsely selected (Meinshausen and Bühlmann, 2010; Hofner et al., 2015), or variables in RF are ranked mistakenly high (Strobl et al., 2007). The most comprehensive approach was used by Ferreira et al. (2021), who employed the RFE for three different learning methods (OLS, Earth, RF), but did not include other prospective learning methods. While all of these studies found variable selection or calculation of variable importance to be a crucial step,

a broader assessment is missing that sheds light on the value of variable selection for different statistical learning approaches. Therefore, we propose to use RFE for our variable selection and suggest three different approaches for computation of the variable ranking to disconnect the variable ranking and the final prediction model.

In this paper we perform a comparative assessment of seven statistical learning models for a comprehensive Austrian data set covering 260 stations. With our study, we specifically addresses the lack of research for comparing these methods in a strongly

seasonal climate with summer and winter low flow regimes. The following research questions will be addressed: (i) How well do statistical learning models perform as compared to established models (and which of the methods perform best)? (ii) What is the effect of different variable preselection methods on the performance of these models? (iii) What is the relative value of non-linear learning models compared to linear ones? The Model performance is evaluated by a repeated nested cross validation (CV) scheme, which provides a confident assessment of how well the models perform at ungauged sites.

## 2 Data

Our study area consists of 260 gauging stations in Austria (Fig. 1). Austria can be described as physiographic and hydrological diverse and is therefore a suitable testbed for regionalisation models. The altitude of gauges ranges from 143 to 1891 m a.s.l.. Annual precipitation vary from 530 mm in the lowlands, up to 2223 mm in high alpine regions. Mean annual temperature ranges from 2 to 11 °C. The 260 gauging stations were all consistently monitored between 1978 and 2013. All these stations





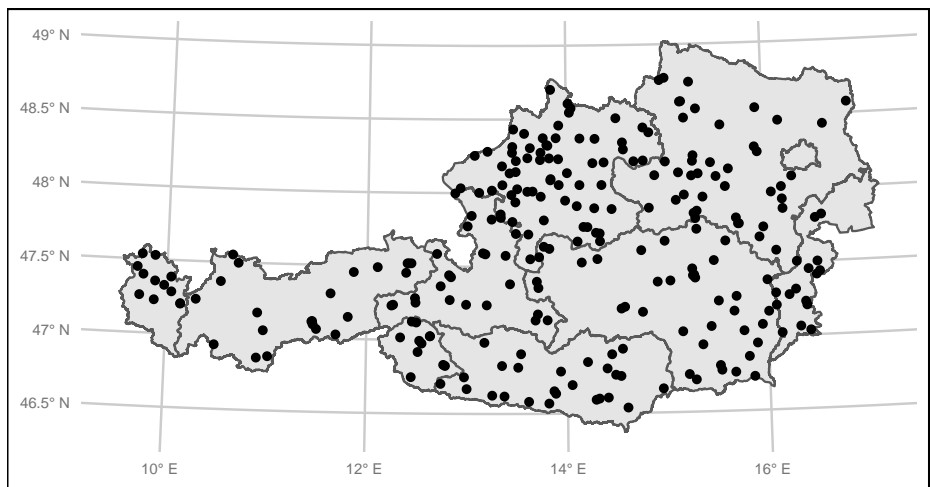

**Figure 1.** Overview of the 260 gauging stations used in the study.

are available at the Austrian Hydrological Service (ehyd) and catchment characteristics were available from previous studies
(e.g. Laaha and Blöschl, 2006). We calculated Q95, where Q95 ($Pr(Q > Q95) = 0.95$) is the flow that is exceeded on 95 % of
all days. Low flow in Austria can be separated into two major seasonal regimes, with low flows in the Alpine region occurring
mainly in the winter half-year and in the lowlands mainly in the summer half-year. To account for these different processes,
we calculated $Q95$ for the summer season (from May to October) and for the winter season (from November to April) and

both low flows (winter and summer) will be analyzed for the full study domain. Summer and winter $Q95$ was subsequently
standardized by the catchment area. The resulting specific low flow discharges $q95$ ($l\,s^{-1}km^{-2}$) were considered in the further
analyzes. For the study area, the average winter low flow is 6.0 $l\,s^{-1}km^{-2}$, which is considerably lower than the summer
low flow, with 8.9 $l\,s^{-1}km^{-2}$ on average. Figure 2 shows that summer $q95$ tends to have more near zero values than winter
$q95$ and additionally summer low flow has a higher variation (standard deviation of 3.2 in the winter and 6.7 $l\,s^{-1}km^{-2}$ in

the summer). Summer $q95$ was transformed by the square root transformation to reach a symmetric distribution. After model
fitting, the predictions were back-transformed for performance evaluation.

## 2.1  Catchment characteristics

We use a set of 87 covariables as possible predictors, some of which are highly correlated. These covariables can be separated
into catchment and climate characteristics. The catchment characteristics used in this study are fully described in e.g. Laaha

and Blöschl (2005, 2006). They consist of nine land use categories, nine geological categories and information about catchment
altitude, stream network density and steepness of the slope in the catchment. An overview is given in Table 1.



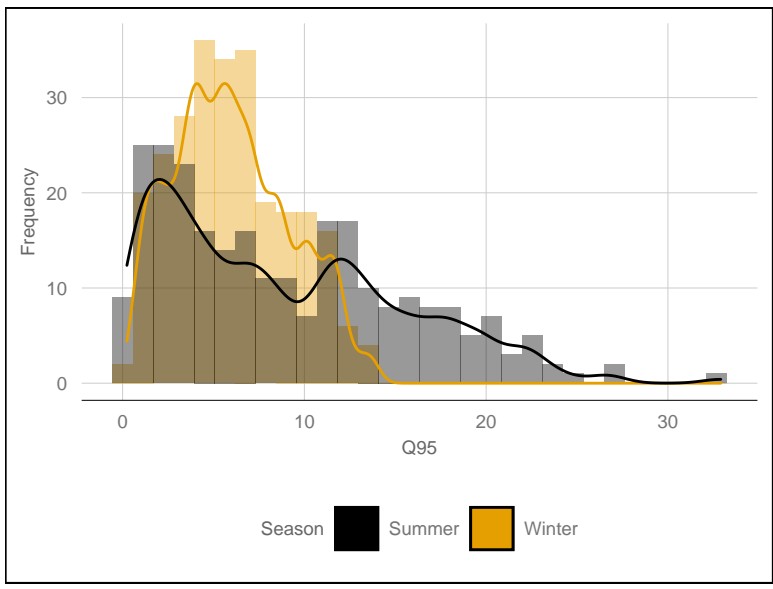

**Figure 2.** Absolute frequency (histogram and kernel density estimate) of summer $q95$ and winter $q95$ for all 260 stations.

## 2.2 Climate characteristics

The calculation of the climate characteristics is based on the the SPARTACUS dataset for daily precipitation (Hiebl and Frei, 2018), daily minimum and maximum temperature (Hiebl and Frei, 2016). Data is available from 1961 to 2018 and the spatial resolution is $1 \times 1$ km. Additionally, we retrieved the HISTALP dataset of the fraction of solid precipitation (Efthymiadis et al., 2006; Chimani et al., 2011), which has a coarse spatial resolution of $5 \times 5$ min and a temporal range from 1801 to 2014. To calculate specific climatological variables for each gauging station, the nearest grid point to the gauging station, which lies inside the catchment was used. The gridded datasets were used to calculate precipitation sums and mean, minimum and maximum temperature. Daily precipitation was further used to estimate the number of days without precipitation for the winter and the summer season. We defined a day without precipitation if the precipitation sum on this day was below 1mm. Potential evapotranspiration was calculated after Hargreaves (Hargreaves, 1994) with the SPEI package in R (Beguería and Vicente-Serrano, 2017). Furthermore, climatic water balance, aridity and the fraction of snow were computed. Snowmelt is approached by a method of Walter et al. (2005), which is included in the R package EcoHydRology (Fuka et al., 2018). All climatological variables were calculated for the period of 1978 to 2013. Our data was restricted up to the year of 2013, as solid precipitation was only available till the end of 2013. For precipitation, climatic water balance, potential evapotranspiration, snowmelt, snow fraction and aridity we calculated average annual sums, and mean sums for each season and for the winter and summer half year (November – April, May – October). Number of days without precipitation were calculated for each half year and averaged over the whole period. Mean, minimum, maximum temperature were calculated for the whole year, and the



| Topographic descriptors | | Geological characteristics | | Landuse characteristics | |
|---|---|---|---|---|---|
| Description | Abbreviation | Description | Abbreviation | Description | Abbreviation |
| minimum, maximum, mean and range of catchment altitude | H.MIN, H.MAX, H.MEAN, H.DIFF | bohemian massif | GEOL.BM | urban | BONU.URB |
| catchment area | area | quaternary sediments | GEOL.QUA | agriculture | BONU.ACK |
| latitude and longitude of gauging stations | latitude, longitude | tertiary sediments | GEOL.TER | permanent crop | BONU.DAU |
| altitude gauging station | altitude | flysch | GEOL.FLY | grassland | BONU.GRU |
| mean catchment slope | M.NEIG | limestone | GEOL.KAL | forest | BONU.WAL |
| percentage of slight slope | SL.FL | crystaline rock | GEOL.KRI | wasteland rocks | BONU.LOS |
| percentage of moderate slope | SL.MG | shallow groundwater table | GEOL.SHAL | wetland | BONU.FEU |
| percentage of steep slope | SL.ST | deep groundwater table | GEOL.DEEP | water surfaces | BONU.WAS |
| stream network density | SDENS | source region | GEOL.QUELL | glacier | BONU.EIS |

**Table 1.** Descriptions of the catchment characteristics that are used in the study. Abbreviations are further used in plots. Landuse characteristics and geological characteristics are given in percentage of the total catchment area.

winter and summer period. Finally, we computed the annual temperature range. All variables related to snow (snowmelt and
snow fraction) were transformed by the square root.

## 3 Methods

This section is divided into two parts. The first part considers the seven statistical learning models used for prediction of summer and winter $q95$. The second part gives a short description of the RFE algorithm and the proposed variable ranking methods. Additionally, there will be an overview of our repeated nested cross validation scheme.



## 3.1 Models

We considered seven statistical learning models that can be structured as follows. Two prediction models are using dimension reduction: (i) principal component regression (PCR) and (ii) sparse partial least squares (sPLS). Additionally, we used two linear models that possess an inherent variable selection method – (iii) the LASSO and (iv) a linear model based boosting approach (GLM). If simple linear terms are not sufficient we can extent the GLM model by non-linear smoothing functions. This results in the (v) generalized additive boosting model (GAM), which has never been applied in a low flow study. Finally, we are using two models that are popular in hydrology: (vi) Random Forest (RF, Tyralis et al. (2019)) and (vii) Support Vector Machine regression (SVM, Raghavendra. N and Deka (2014)).

All models can be considered as regression models where the response variable $Y$ is a vector of length $N$ ($N = 260$) catchment observations, which can either be summer $q95$ or winter $q95$. The predictor matrix $X$ is a $N \times p$ matrix with elements $x_{ij}$ representing the values of $p = 87$ numeric predictors for the $i$-th catchment.

### 3.1.1 LASSO

The LASSO was originally introduced by Tibshirani (1996), where the regression coefficients $\beta^{lasso}$ can be defined as follows:

$$\beta^{lasso} = argmin_\beta \{1/2 \sum_{i=1}^{N}(y_i - \beta_0 - \sum_{j=1}^{p} x_{ij}\beta_j) + \lambda \sum_{j=1}^{p}|\beta_j|\}, \tag{1}$$

with $\beta_0$ as the intercept and $\beta_j$ as the regression coefficients. The LASSO model performs a penalized model optimization known as L1 regularization that reduces parameters and shrinks the model. A tuning parameter lambda controls the strength of the penalty and thus the parsimony of the model. Setting lambda to zero results in the ordinary least squares regression estimates, whereas large values of lambda lead to a simple intercept model. In between these limits, lambda is performing a continuous subset selection (Hastie et al., 2009). We are using the glmnet package in R for computation (Simon et al., 2011), where the coefficients are estimated by cyclical coordinate descent (Friedman et al., 2010). An optimal solution for lambda is chosen by 10-fold cross validation, where we choose lambda by the 1-standard error rule. We prefer this over using lambda with minimum error, as this results in sparser and more robust models in our case. The lasso approach can handle high correlated data, coefficients can be shrunk to zero and correlated variables do not enter the final model (Friedman et al., 2010).

### 3.1.2 Principal component regression (PCR)

Principal component regression (PCR) is a regression method that can deal with multicollinearity and high dimensional data. PCR projects the predictor matrix $X$ on an orthogonal space, which ensures that the final predictors are uncorrelated. The final dimension of our regression problem can thus be reduced from $p$ (number of predictors) to $M$ (number of principal components). Using $M = p$ would result in the least squares estimate for the full parameter space. The principal components of $X$ are defined as $z_m = X\nu_m$, where $z_m$ are the principal components and $\nu_m$ are the principal directions of X. The final





regression coefficients of the principal components ($z_m$) can be defined by (Hastie et al., 2009):

$$\beta^{pcr}(M) = \sum_{m=1}^{M} \theta_m \nu_m, \tag{2}$$

where $\theta_m$ is $\theta_m = Cov(z_m, y)/Var(z_m)$. As for the LASSO, $X$ has to be standardized in PCR before estimating the regression coefficients, as the principal components are dependent on the scaling of the initial variables. The number of principal components M is optimized by a 10-fold cross validation and the regression coefficients are estimated by ordinary least squares.

We fit our PCR model using the pls package in R (Mevik et al., 2020).

### 3.1.3  Sparse partial least squares regression (sPLS)

Additionally to LASSO and PCR we propose a third dimension reduction method, partial least squares regression (PLS). PLS uses linear combinations of $X$ for the regression of Y, but these linear combinations are now constructed in dependence to Y (Hastie et al., 2009). This overcomes the drawback of PCR, which can not guarantee that the first principal components of $X$

are most suited to predict Y. PLS, originally developed by Wold (1966) is an iterative process that starts (i) with centering the response variable ($U_1$) and the predictor variables ($V_{1j}$). Next (ii), $p$ univariate regression models are constructed by regressing $U_1$ against each centered predictor variable $V_{1,j}$, which gives us $p$ regression coefficients. These regression coefficients are now used to compute the first PLS component, which is the weighted average (iii) defined as:

$$T_1 = \sum_{j=1}^{p} w_j b_j V_{1j}, \tag{3}$$

where $b_j$ are the univariate regression coefficients defined as $b_j = Cov(V_{1j}, U_1)/Var(V_{1j})$, and the weights are $w_i = Var(V_{1j})$. In a next step (iv) $U_2$ is estimated by computing the residuals of the regression model $U_1 \sim T_1$. Furthermore, the predictor matrix $V_{2j}$ is updated by the residuals of the models $V_{1j} \sim T_1$, where each predictor variable is regressed against the first PLS component. This process is repeated until M PLS components are extracted (de Jong, 1993). As PLS yields a high variability in the performance evaluation, we used an adapted PLS procedure, which is called sparse PLS (sPLS). This method was in-

troduced by Chun and Keleş (2010) and includes a variable selection, as a $L1$ penalty is added to the calculation of the PLS components. The model is tuned by 10-fold CV in the spls package in R (Chung et al., 2019).

### 3.1.4  Linear and non-linear model based boosting

In this section the model based boosting algorithm is presented, which is used for fitting a simple linear model (GLM) and a generalized additive model (GAM). Boosting refers to an ensemble learning approach that converts a set of weak models,

termed learners, into a strong model with better model fit. A current approach is functional gradient descent boosting, a stage-wise, additive approach, which improves a fitted model by adding, each step, a new learner that reduces the model errors. When predictors $X$ are entered as separate learners, so that the prediction function $f$ is an additive estimate based on simple linear terms $f_j(x_j) = \beta_j x_j$, the approach allows one to obtain an inherent variable selection and can penalize regression coefficients





(Mayr and Hofner, 2018). In addition, model based boosting can deal with multicollinearity and can handle e.g. linear, non-
linear, spatial or random effects (Hofner et al., 2014).

Model based boosting, as applied in this study, aims to minimize an empirical risk

$$R := 1/N \sum_{i=1}^{N} \rho(y_i, f_i) \tag{4}$$

based on the so-called loss function $\rho(y, f)$ characterizing the inadequacy of the fitted model. For regression problems including
GLM and GAM we use the squared error loss function

$$\rho(y, f) = \frac{1}{2}|y - f|^2, \tag{5}$$

which results in a stage-wise least-square minimization of the residuals. The boosting algorithm is an iterative process, with
the following steps (Bühlmann and Hothorn, 2007; Mayr and Hofner, 2018; Melcher et al., 2017):

1. In a first step all base learners are defined. A base learner can be an e.g. linear, non-linear, spatial or random effect.
   The two models used in this study incorporate linear base learners for the linear model (GLM), and linear and non-linear
effects for the GAM model. As shown by initial analysis, spatial effects or higher order interaction effects did not improve
   the prediction performance, hence they were discarded from the analysis. Non-linear effects are modeled as P-splines
   (Schmid and Hothorn, 2008), which are decomposed into an unpenalized linear base learner and a penalized non-linear
   base learner, each with 1 degree of freedom. The non-linear base learner is centered by subtracting the unpenalized linear
   part. This approach is proposed by Kneib et al. (2009) and Fahrmeir et al. (2004) and offers the possibility to spot the
predictor variables that are added as linear or non-linear effects. This leads to $p + 1$ ($p$ predictor variables plus one term
   for an intercept) linear base learners for the GLM model, and $2p + 1$ base learners for the GAM model.

2. In the first iteration the counter $m$, which is the number of boosting steps, is set to 0 and the initial function estimate is
   set to $\hat{f}^{[m]} = \hat{f}^{[0]}$. The first function estimate ($\hat{f}^{[0]}$) is determined by an offset, which is the mean of the response for our
   purpose ($\hat{f}^{[0]} := \overline{y}$).

3. The following steps are now repeated until the maximum number of boosting steps is reached, which was fixed to 1000
   in this study:

   – The tuning parameter $m$ is increased by 1.

   – The negative gradient $-\frac{d(\rho)}{d(f)}$ is computed and evaluated at the function estimate of the previous iteration $f^{[m-1]}$,
     resulting in the negative gradient vector $u^{[m]}$.

– Each base learner is now fitted by univariate regression against $u^{[m]}$ and the best fitting base learner ($=: \hat{g}^{[m]}$) is
     selected.

   – The function estimate is updated by: $\hat{f}^{[m]} := \hat{f}^{[m-1]} + \nu\hat{g}^{[m]}$, where $\nu$ is a value between 0 and 1 and if $\nu$ is
     sufficiently low, the risk of finding only a local minimum is reduced. Therefore, $\nu$ was set to 0.1 in this study.



In each boosting step ($m > 0$) only one base learner is selected and can be chosen again in later iterations. The number of
boosting steps are optimized by a 10-fold CV. Although studies indicate that repeated CV would yield more robust results
(Seibold et al., 2018), due to computational costs and the low risk of overfitting 10-fold CV seemed sufficient. The model
boosting was performed using the mboost package in R (Hothorn et al., 2021).

### 3.1.5 Random Forest (RF)

Random Forest (RF) is a bagging (bootstrap aggregating) method originally developed by Breiman (2001). In a RF model
multiple regression trees are generated using bootstrap samples and their predictions are averaged to yield the RF estimate.
Bootstrapping decorrelates the individual trees and adds some randomness to the predictions. There are several packages in
R that can estimate RF models, but due to the computational burden of the study we used the fast ranger package (Wright
and Ziegler, 2017). An RF model has several hyperparameters that can be tuned. The number of trees used for bagging is one
of these parameters, but it just has to be sufficiently high, as more trees do normally not impair the prediction performance.
In this study we used 500 trees. Another parameter that needs to be optimized is the size of each bootstrap sample. We used
a grid search from 0.7 to 0.9 for finding an optimal sample size for each bootstrap sample. Sampling was applied without
replacement. Further, the number of variables that are randomly chosen for each split needs to be set, which was determined
by $p/3$. We used the estimated response variance as splitting criterion because rank-based approaches or the use of extremely
randomized trees (Geurts et al., 2006) did not improve prediction accuracy.

### 240   3.1.6 Support Vector Machines (SVM)

Support Vector Machines have their origin in classification but can be extended to regression problems. The method in its basic
form uses a training dataset to create a line (or hyperplane) that separates the data into classes. The support vectors are the
data points closest to the line or hyperplane and have the most influence on parameter estimation. In SVM regression each
of the predictor variables can be transformed to a set of basis function $h_m(x_j)$. Hence the regression function $f(x)$ can be
approximated by (Hastie et al., 2009):

$$f(x) = \sum_{m=1}^{M} \beta_m h_m(x_j) + \beta_0. \tag{6}$$

The number of M basis function is not limited and to estimate the coefficients $\beta_0$ and $\beta_m$, $H(\beta, \beta_0)$ has to be minimized:

$$H(\beta, \beta_0) = \sum_{i=1}^{N} V(y_i - f(x_i)) + \lambda/2 \sum \beta_m^2. \tag{7}$$

$V$ can be any loss function, but in the initial idea of Vapnik (2000), it is defined by a threshold $r$. If the residuals are higher than
this value, they are included in the penalization of the model, if the residuals are lower they are discarded (Worland et al., 2018).
Hence, the SVM regression is insensitive to outliers, as coefficients will be penalized in respect to them. The fast computation
of the coefficients is achieved by only computing the inner product of each $x_j$. Therefore, different kernels can be used and we





decided to use the non-linear radial kernel in our study. The SVM regression model was estimated by the e1071 package in R (Meyer et al., 2021).

## 3.2 Variable preselection for parsimonious models

The variable selection procedure of this study is based on the RFE algorithm. RFE is a prospective method, that initially ranks the predictor variables after some measurements of importance and the least important variables are removed in a backward procedure (Granitto et al., 2006). The final number of variables are determined by an error measurement of an independent test set. In this part we will present our three approaches for variable ranking, the error measurement to define the number of variables and a short overview of the validation scheme.

### 3.2.1 Variable ranking methods

We test three different methods for the variable ranking of the RFE. Thus, we can differentiate between the prediction accuracy of the prediction models and the capability of different variable ranking methods for producing parsimonious models.

- The first variable ranking method is the lasso ($lasso_{rank}$), which is applied as described in Sect. 3.1.1, except that standardized coefficients are calculated.

- Second, we use a linear model based boosting approach ($glm_{rank}$). Every model is estimated by 500 boosting steps, since boosting shows only slow overfitting behavior (Fig.3). One disadvantage is that some non-influential variables will be ranked, but computation time is reduced. For calculation of the standardized coefficients we apply the variable importance function of the caret package in R (Kuhn, 2021).

- The third method ($cf_{rank}$) uses conditional forests (Hothorn et al. (2006); Strobl et al. (2007, 2009)) for variable ranking, where the standardized coefficients are a sum of the main and interaction effects for each variable. Standardized coefficients are again calculated using the variable importance function of the caret package.

Our variable ranking is computed over a bootstrap sample to improve its robustness with respect to data. Therefore, an initial dataset $D$ is split up into 25 bootstrap samples ($B = (b_1, b_2, b_3, ..., b_{25})$). Sampling is performed without replacement and a sample size of 68.2 % is used.

In a next step, each of the bootstrap samples is fitted to one of the variable ranking methods. For each bootstrap sample and each method standardized coefficients denoted as $\beta_{j,b}^{method}$ are returned, where $b$ refers to the bootstrap sample, $method$ the variable ranking method ($lasso_{rank}, glm_{rank}, cf_{rank}$), and $j$ is the considered predictor. The variable importance of each bootstrap sample for each selected coefficient is calculated by:

$$varimp_{j,b}^{method} = \beta_{j,b}^{method} \cdot 100 / \sum_{j=1}^{p} \beta_{j,b}^{method}. \tag{8}$$





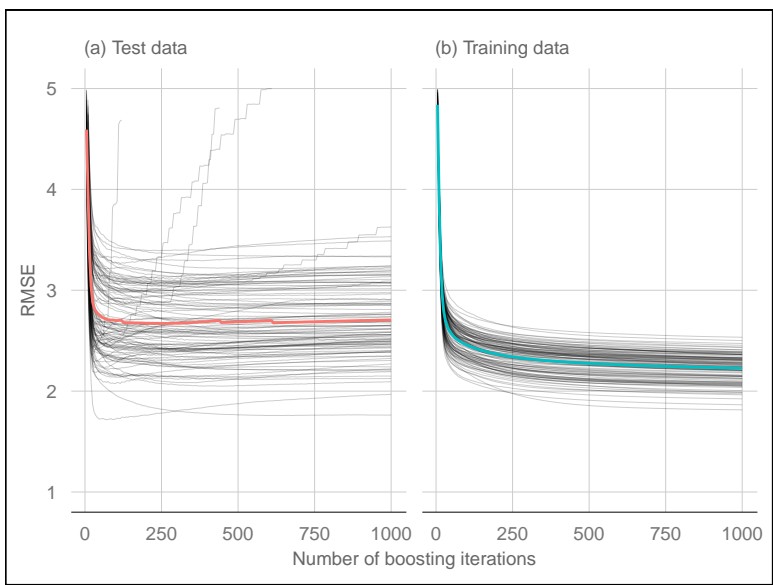

**Figure 3.** Prediction error for the unseen test data (left panel) and the training data (right panel). Bootstrapping was performed 100-times where the training set consists of 70 % of the observations.

Final variable rankings are computed by averaging over all 25 bootstrap samples (Eq. 9). The variables are ranked from highest to lowest.

$$varrank_j^{method} = rank(1/25 \sum_{i=1}^{b} varimp_{j,i}^{method}). \tag{9}$$

Variables that were never selected in the 25 boostrap samples are not considered for the variable selection. This yields $n_{var}$
285  preselected variables for each method.

### 3.2.2  Model specific preselection of variables

The variable rankings are now calculated for each of the seven prediction models. Each prediction model is fitted to the best $l$
($l = 2, 3, ..., n_{var}$) ranked variables, and for each prediction vector $\hat{y}_i$ the root mean squared error ($RMSE$) as a measure of fit
is calculated:

290  $$RMSE = \sqrt{1/N \sum_{i=1}^{N} (y_i - \hat{y}_i)^2} \tag{10}$$

As one aim of this study is to get unbiased predictions (Solomatine and Ostfeld, 2008; Ambroise and McLachlan, 2002), we employ a nested CV scheme where a inner (10-fold) CV loop is used for variable selection and model tuning, and a outer (10-fold) CV loop is used for evaluating the predictive performance of the final models. This should make the evaluation more independent from model fitting. The $RMSE_l$ for each candidate variable selection $l$ is then simply the average over the 10
295  folds of the inner CV.





The final number of variables ($n_{final}$) of a method is then obtained from the relationship of the $RMSE_l$ versus the number of variables $l$, as exemplified in Fig. 4. A common option is to use the $\min(RMSE)$ for determining $n_{final}$, but this would only yield to a small reduction in the number of variables. We therefore propose to use a somewhat (+5 %) higher residual error, i.e. $1.05 \times \min(RMSE)$, as reference point. This should make the models more parsimonious with only a slight loss in performance.

In a final step, we determine the specific variables that are used for predictions of the outer loop of the nested CV. For this we use all obtained variable rankings of each of the 10 inner folds of the CV run and average each variable ranking over these 10 folds. The best $n_{final}$ variables of these final rankings are used for prediction. This nested CV is repeated 10-times for summer $q95$ and winter $q95$.

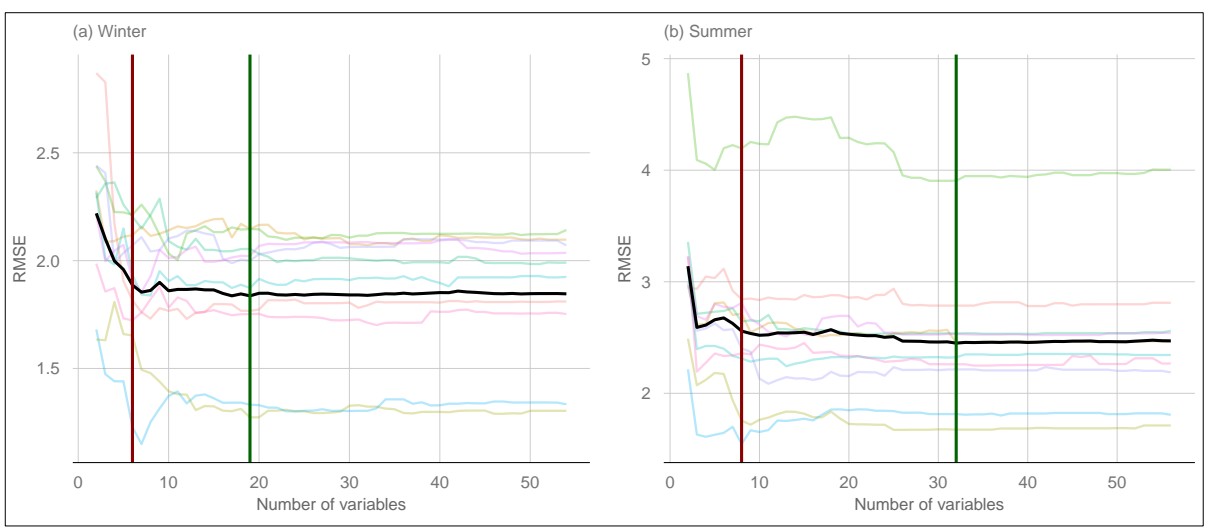

**Figure 4.** Variable selection for a GAM boosting model based on $RMSE$ graphs. The red vertical line indicates the number of variables selected by a 5 % increase of the $\min(RMSE)$, the green line is the number of variables obtained by $\min(RMSE)$. The Black graph is the average over 10 inner CV-folds, and coloured graphs represent individual folds.

### 3.2.3 Performance evaluation metrics

Model evaluation was performed by the root mean squared error $RMSE_{CV}$ and the relative root mean squared error $RRMSE_{CV}$ of each CV repetition:

$$RRMSE_{CV} = RMSE_{CV}/\overline{y}, \tag{11}$$

where $\overline{y}$ is the mean of all observations. We further compare our results by the cross validated $R^2_{CV}$ defined as:

$$R^2_{CV} = 1 - \frac{RMSE^2}{1/N \sum_{i=1}^{N}(y_i - \overline{y})^2}. \tag{12}$$





For a more focused assessment of individual catchments in terms of how CV performance depends on climate and catchment characteristics we use the absolute normalized error $ANE_{CV}$ of the $i$-th catchment:

$$ANE_{CV,i} = |(\hat{y}_i - y_i)|/y_i. \tag{13}$$

## 4 Results

### 315 4.1 Model Performance without variable preselection

Figure 5 shows the performance of all models without variable preselection and Table 2 presents the performance metrics of all models. The best performing model for winter low flow is the SVM model with a median $R^2_{CV}$ of 0.70 over all 10 CV-runs. It is followed by the GLM (0.69) and RF (0.68), and a group of similar performing models (LASSO, GAM, PCR and sPLS) with an $R^2_{CV}$ of 0.66. Summer low flow is generally reaching a higher prediction accuracy, with a $R^2_{CV}$ of 0.86 (GAM) and 320 0.85 (GLM) for the two boosting approaches, and a somewhat lower performance for the LASSO, sPLS, RF, SVM (all 0.84) and the PCR (0.83).

Additional insights can be gained by stratifying the predictions by specific low flow magnitude into three parts; the first part

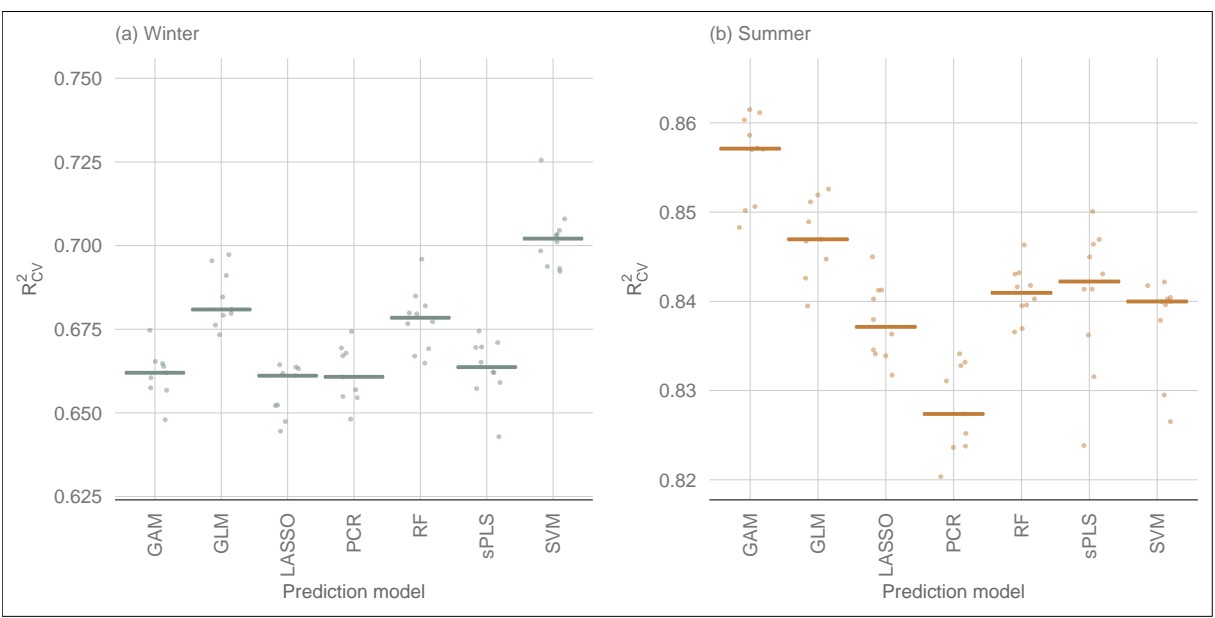

**Figure 5.** Performance of 10 CV-runs of statistical learning models without variable preselection. Horizontal lines display the median of the CV-runs.

containing observations smaller than the first quartile, the second part ranging between the first and the third quartile and the third part considering only the observations higher than the third quartile. For each of the three parts we calculated the 325 $RRMSE_{CV}$. High $q95$ winter values are reaching similar performances for all models with a $RRMSE_{CV}$ of 0.24 to 0.25.





Middle ranged values are best approximated by the LASSO model ($RMSE_{rel} = 0.24$), followed by the RF and the SVM (both 0.25) and slightly worse performances by the two boosting models (0.26) and sPLS and PCR (0.27). The main differences between the model predictions can be observed for the lowest observation class, where the SVM model with a $RRMSE_{CV}$ of 0.58 performs substantially better than the other models. The next best model in this class is the GLM (0.66), and models such as RF (0.75) and GAM (0.78) have less prediction accuracy.

For summer low flow, low $q95$ values again have a higher relative error than medium or high $q95$ values. For the lowest observation quartile, the GLM shows the best performance ($RRMSE_{CV}$ of 0.49), compared to most other models ($RRMSE_{CV}$ between 0.54 and 0.56). The sPLS (0.62) and the RF (0.71) show a much lower prediction accuracy in this class. Differences for moderate summer $q95$ values are only marginal with a range of 0.3 (SVM) to 0.32 (PCR). High summer $q95$ values are somewhat better approached by the GAM (0.2), with a slightly higher $RRMSE_{CV}$ of all other models ranging from 0.21 to 0.23.

### 4.2 Effect of variable preselection

The use of variable preselection can significantly reduce the complexity of all models with only a small loss of performance (Fig. 6). For winter low flow, variable selection leads to a median $R^2_{CV}$ decrease of 5.1 % ($cf_{rank}$), 5.8 % ($glm_{rank}$) and 7.1 % ($lasso_{rank}$) over all models. The spread in performance across models is a bit higher for the $lasso_{rank}$ with an interquartile range (IQR) of 4.1 %, compared to an IQR of 3 % for $cf_{rank}$ and 2.8 % for $glm_{rank}$. Although $cf_{rank}$ yields a slightly better performance than $glm_{rank}$ and $lasso_{rank}$, the conditional forest approach requires 35 variables, where $glm_{rank}$ and $lasso_{rank}$ are only using 12 and 14 variables for winter $q95$. The $glm_{rank}$ and $lasso_{rank}$ are therefore much more effective. The number of variables for the three variable ranking methods have almost no dispersion, with a IQR of 1 ($glm_{rank}$), 2 ($cf_{rank}$) and 3.5 ($lasso_{rank}$). Interestingly, the highest performance loss is observed with the linear boosting model (GLM) for all three variable ranking methods. This is due to the nature of boosting methods, whose main strength is efficient parameter estimation for high-dimensional multicollinear data sets. Clearly, variable preselection affects the performance of the method.

In contrast to winter low flow, where the performance loss corresponds well with the +5 % residual error specification, variable selection for summer low flow only leads to a minor loss in performance (Fig. 6b). Here, the median $R^2_{CV}$ decrease is only 1 % for $cf_{rank}$ and $lasso_{rank}$, and 0.8 % for the $glm_{rank}$. Also the differences between models are very small. Again, $cf_{rank}$ yields a substantially higher number of variables (22), than the $glm_{rank}$ method (8 variables) and the $lasso_{rank}$ (9 variables). Furthermore, the IQR of the number of variables shows that the selected number of variables is about the same in all models for $lasso_{rank}$ (IQR = 3) and $glm_{rank}$ (IQR = 2), but greatly differs between the models based on $cf_{rank}$ (IQR = 23). This spread reflects a much lower parameter-reduction efficiency of $cf_{rank}$ for the linear models (22 predictors for PCR, 35 for LASSO, 42 for sPLS, 50 for GLM) than for the non-linear models (12 predictors for RF, 17 for SVM, 18 for GAM). Among all models, the RF provides the most parsimonious model for summer low flows. It consists (on median) of only three variables when $lasso_{rank}$ and $glm_{rank}$ are used for variable selection, and has a performance loss of less than 1 %.


| Model | Variable ranking | Winter | | | Summer | | |
|---|---|---|---|---|---|---|---|
| | | median RMSE | sd RMSE | median $R^2_{CV}$ | median RMSE | sd RMSE | median $R^2_{CV}$ |
| GAM | $cf_{rank}$ | 1.95 | 0.024 | 0.622 | 2.65 | 0.022 | 0.844 |
| GAM | $glm_{rank}$ | 1.94 | 0.041 | 0.626 | 2.67 | 0.068 | 0.842 |
| GAM | $lasso_{rank}$ | 1.99 | 0.033 | 0.603 | 2.67 | 0.052 | 0.842 |
| GAM | none | 1.84 | 0.109 | 0.661 | 2.53 | 0.041 | 0.857 |
| GLM | $cf_{rank}$ | 1.90 | 0.033 | 0.639 | 2.78 | 0.038 | 0.828 |
| GLM | $glm_{rank}$ | 1.93 | 0.021 | 0.637 | 2.78 | 0.067 | 0.828 |
| GLM | $lasso_{rank}$ | 1.98 | 0.024 | 0.609 | 2.77 | 0.043 | 0.829 |
| GLM | none | 1.79 | 0.328 | 0.69 | 2.62 | 0.216 | 0.847 |
| LASSO | $cf_{rank}$ | 1.95 | 0.024 | 0.621 | 2.76 | 0.023 | 0.832 |
| LASSO | $glm_{rank}$ | 1.97 | 0.02 | 0.611 | 2.73 | 0.033 | 0.835 |
| LASSO | $lasso_{rank}$ | 1.97 | 0.015 | 0.613 | 2.76 | 0.027 | 0.832 |
| LASSO | none | 1.85 | 0.107 | 0.657 | 2.71 | 0.036 | 0.837 |
| PCR | $cf_{rank}$ | 1.93 | 0.03 | 0.627 | 2.85 | 0.045 | 0.819 |
| PCR | $glm_{rank}$ | 1.93 | 0.026 | 0.629 | 2.8 | 0.075 | 0.825 |
| PCR | $lasso_{rank}$ | 1.96 | 0.021 | 0.616 | 2.8 | 0.04 | 0.826 |
| PCR | none | 1.85 | 0.748 | 0.659 | 2.79 | 0.563 | 0.826 |
| sPLS | $cf_{rank}$ | 1.9 | 0.038 | 0.641 | 2.81 | 0.037 | 0.825 |
| sPLS | $glm_{rank}$ | 1.94 | 0.026 | 0.625 | 2.8 | 0.8 | 0.825 |
| sPLS | $lasso_{rank}$ | 1.96 | 0.023 | 0.615 | 2.79 | 0.31 | 0.826 |
| sPLS | none | 1.84 | 0.024 | 0.664 | 2.66 | 0.067 | 0.842 |
| RF | $cf_{rank}$ | 1.86 | 0.04 | 0.656 | 2.69 | 0.044 | 0.839 |
| RF | $glm_{rank}$ | 1.89 | 0.032 | 0.642 | 2.73 | 0.022 | 0.835 |
| RF | $lasso_{rank}$ | 1.85 | 0.03 | 0.657 | 2.73 | 0.054 | 0.834 |
| RF | none | 1.8 | 0.026 | 0.678 | 2.68 | 0.25 | 0.841 |
| SVM | $cf_{rank}$ | 1.75 | 0.048 | 0.694 | 2.67 | 0.04 | 0.842 |
| SVM | $glm_{rank}$ | 1.86 | 0.034 | 0.654 | 2.7 | 0.037 | 0.839 |
| SVM | $lasso_{rank}$ | 1.85 | 0.05 | 0.659 | 2.7 | 0.033 | 0.839 |
| SVM | none | 1.73 | 0.029 | 0.702 | 2.69 | 0.043 | 0.84 |

**Table 2.** Overview of the prediction performance of all models. The median and standard deviation (sd) are calculated over the 10 CV-runs for each model.







**Figure 6.** Reduction of $R^2_{CV}$ is computed in respect to the model performance without any variable preselection. Each point is the median number of variables of one CV-run and one prediction model and the related $R^2_{CV}$. Each line shows (horizontal = Number of variables, vertical = Reduction in $R^2_{CV}$) the upper and lower quartile for a variable ranking method. Colours can be found in Ram and Wickham (2018)

### 4.3 Importance of predictors (from variable rankings)

We performed variable rankings for each of the three ranking methods 1000 times inside the CV runs. In this section we discuss the 10 best ranked variables for each variable ranking method, defined by the average rank over all 1000 repetitions. We focus on the two linear ranking methods, as the non-linear method $cf_{rank}$ did not perform well.

Figure 7 gives an overview of the rank counts for winter low flow and shows that the catchment altitude is on average the highest ranked variable. Meteorological based variables appear four ($glm_{rank}$) and five ($lasso_{rank}$) times under the ten best ranked

variables. Predictors rated by both methods are aridity and snowmelt in the winter months. The $glm_{rank}$ lists preconditions as precipitation or dry days in the summer as important variables. However, $lasso_{rank}$ found dry days in winter and over the





whole year to be more important. Geological variables as percentage of Quaternary sediments or limestone enter the models as indicator for catchment processes. Finally, land use characteristics as proportion of agriculture area and wasteland rocks were found for the $glm_{rank}$, where only the fraction of grassland is rated by $lasso_{rank}$. Both are correlated with the proportion of

lowland/high mountain areas and can be interpreted as topological characteristics as well.

A slightly different picture emerges from the assessment of summer low flow (Fig. 8), where the three highest ranked variables are maximum catchment altitude, mean catchment slope and dry days in summer for both variable ranking methods. Topological descriptors play a somewhat more dominant role for summer low flow, as four ($glm_{rank}$) and five ($lasso_{rank}$) variables are highly ranked. Apart from mean catchment slope and maximum catchment altitude, difference in catchment altitude and

stream network density are also found for both variable ranking methods. Two meteorological variables - aridity in winter (both methods) or autumn ($lasso_{rank}$) and the annual temperature range ($lasso_{rank}$) - are found by each of the methods for summer $q95$ additionally to the dry days in summer. Finally, geological features as the proportion of Flysch, or land use variables as the fraction of grassland are highly ranked for both approaches.

## 5 Discussion

### 5.1 Predictive performance and benchmarking

We showed that statistical learning models can yield high prediction accuracy. It is now interesting to assess how the models fit into the picture of existing national and international studies. We first assess the performance relative to low flow regionalization studies for the Austrian study area. Laaha and Blöschl (2006) fitted a multiple regression model to annual low flow $q95$ on 325 (sub-)catchments. They reported a performance of $R^2_{CV} = 0.70$ when the study area was subdivided into regions that exhibit

similar seasonal characteristics of low flow. In a subsequent study, Laaha et al. (2014) showed that Top-Kriging can outperform regional regression, especially when interpolating between gauges at the larger rivers. Top-kriging (TK, Skøien et al., 2006) is a geostatistical method that uses stream-network distance for low flow prediction and was shown to be more adequate than ordinary kriging approaches. For comparison of our results with the current benchmark (TK), we used the same 10-fold CV runs as for our statistical learning models. TK yields a median $R^2_{CV}$ of 0.68 for winter low flow, which is slightly below the

SVM and the GLM model and equivalent to the RF model. TK is performing similar to most models for summer $q95$ with a median $R^2_{CV}$ of 0.84, and performs quite similar as the two boosting approaches. Hence, we show that statistical learning models can perform as good or even better as the current benchmark TK for summer and winter low flow in the Austria study area.

It is also interesting to compare our findings to existing studies that assess statistical learning methods for low flow estimation.

However, comparison of performance metrics across studies is not straightforward. Worland et al. (2018) and Ferreira et al. (2021) assessed their prediction models to low flow characteristics such as $Q95$ and a quite similar characteristic $7Q10$, but these were not standardized by catchment area as in our study. This can lead to superior performance metrics, particularly if there are significant variations in catchment size within the sample. Worland et al. (2018) reported a NSE (which is equivalent to the $R^2_{CV}$ in this study) of 0.92 for the meta cubist model, and Ferreira et al. (2021) reported a NSE value of almost 1. However,





**Figure 7.** Count of all variable rankings of all 1000-iterations on a log-scale for winter low flow. The ten best variables are listed after their averaged rank.

the scatterplots of the studies suggest that errors are still considerable, especially for the low observation values. Although the studies are not directly comparable to our study in terms of performance, a qualitative comparison is still warranted. Both studies found tree based methods, including the meta cubist model (Worland et al., 2018), the RF (Zhang et al., 2018), and tree based boosting (Tyralis et al., 2021) having a higher prediction accuracy than the other models. Our study complements existing studies by examining additional learning models. Our results suggest that the SVM and the GAM boosting model

can outperform tree-based models for the Austrian setting. However, differences in performance are rather small, so that other methods (e.g. GLM, LASSO, and tree-based RF) can also be considered well suited.

One major research gap addressed by this study is the separate evaluation of statistical learning models for seasonal low flow







**Figure 8.** Count of all variable rankings of all 1000-iterations on a log-scale for summer low flow. The ten best variables are listed after their averaged rank.

processes. All statistical learning models of this paper can be classified as global models, as all gauges are considered in the same model without catchment grouping. Earlier studies showed that regional regression can increase the prediction accuracy

compared to global regression (Laaha and Blöschl, 2006, 2007) as different low flow generation processes apply to summer and winter regions. Here we pursue a different strategy in which we separate summer and winter processes by a temporal stratification into summer and winter low flows. We found that analysing winter and summer low flow indices individually leads to increased prediction accuracy, especially for summer $q95$. This emphasizes that prediction accuracy of a specific model is influenced by the underlying hydrological process, and different models can be suitable for different applications

(Worland et al., 2018). For comparison, preliminary results without consideration of the seasonal regime of our study area has





led to a prediction accuracy of a median ($R^2_{CV}$) of 0.66 to 0.74 which is very similar to the earlier studies on annual low flow (TK 0.75).

## 5.2 Predictive performance as a function of catchment characteristics


In a comparative assessment on low flow studies based on the PUB assessment report (Blöschl et al., 2013), Salinas et al. (2013), showed that prediction accuracy is not only a function of model selection, but of the specific setting of the study area. The assessment did not contain statistical learning models, so we want to embed our results in their findings. Figure 9, which is equivalent to Fig. 5 of Salinas et al. (2013), shows the $ANE_{CV}$ as a function of the aridity index, the catchment area and the catchment altitude. Our study confirms the finding of the PUB assessment report that the prediction accuracy decreases as the aridity of the study area increases (Salinas et al., 2013). Although stations with an aridity index over one are missing, the trend


is clearly evident. No trend is evident for the winter low flows, which are more driven by freezing processes than by a climatic water balance deficit. Decreasing performance in arid regions for drought detection was also found by Haslinger et al. (2014). This effect may be additionally intensified because in arid regions the mean of observations can be near zero.

Another hypotheses of Salinas et al. (2013) is that a higher elevation increases the prediction accuracy. In this context we found remarkably divergent results for summer and winter low flow. Whereas our findings for summer low flow are in line


with Salinas et al. (2013), we could not identify a clear tendency for winter low flow. Catchments located in lowlands and mountainous areas have a somewhat larger $ANE_{CV}$ than catchments with an elevation between 450 and 1500 meters. This suggests that winter low flows are better predictable in colder mid-mountain catchments than in warmer lowland catchments, where occurrence of frost events varies from year to year. Finally, we can show that prediction accuracy is increasing with catchment size, which is fully consistent with Salinas et al. (2013).


Another finding of (Salinas et al., 2013) that is not captured by Fig. 9 is that predictions of low flows in cold climates are reaching a lower prediction accuracy than in humid, thus warmer climates. A comparable effect can be observed when comparing the results for winter low flow and summer low flow, where the best performing model in winter has a $R^2_{CV}$ of 0.70 and in summer 0.86. This divergence can be explained by the more complex hydrological processes of winter low flows (Salinas et al., 2013). It is shown here that this performance gap applies to seasonal climates with a warm season and a frost period in


the same way as between cold and humid climates.

## 5.3 Linear vs. non-linear models

All studies that conducted a comparative assessment of statistical learning models for low flow estimation highlighted that non-linear models are superior in respect to linear approaches (e.g. Worland et al., 2018; Ferreira et al., 2021; Zhang et al., 2018; Tyralis et al., 2021). In principle this is consistent with our findings, where winter low flow is best predicted by the SVM


model, and summer $q95$ is best approached by the GAM model. However, we showed that linear statistical learning models such as GLM or sPLS perform almost as well in our study. To shed more light on this issue we assessed the relative value of the GAM over the GLM boosting model in more detail. Both models are equivalent in case of linear relationships, but the GAM offers the possibility to extend the GLM with non-linear relationships if these improve the model. The comparison yields



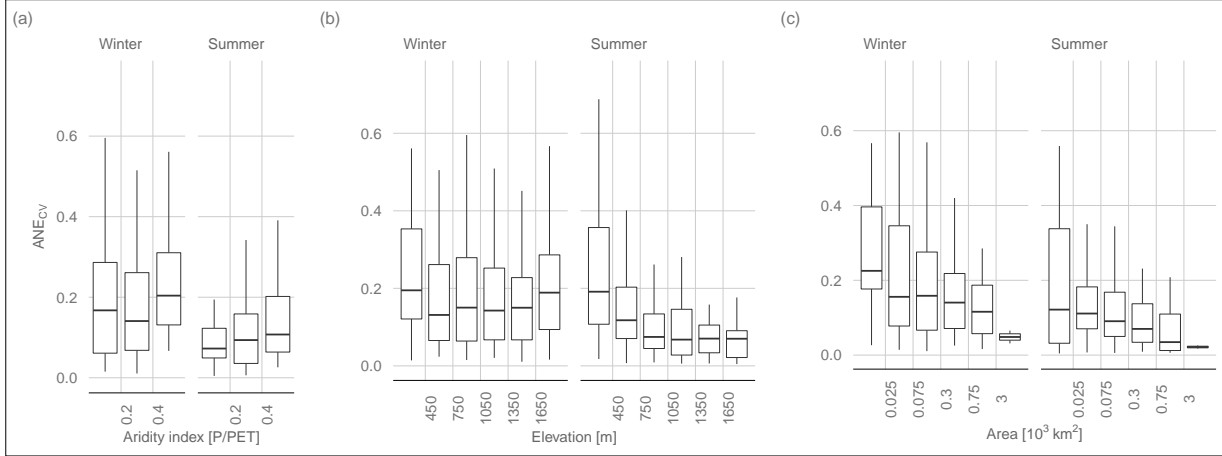

**Figure 9.** $ANE_{CV}$ for summer and winter $q95$ as a function of the aridity index (a), elevation (b), and catchment area (c). Only the best performing models are shown (winter SVM, summer GAM, both without variable preselection). The boxplots summarize the $ANE_{CV}$ averaged over all 10 CV-runs for each of the 260 stations.

that the GAM is selecting additional non-linear effects to increase the goodness-of-fit. However, the additional effects do not
increase the predictive performance of model. In fact, the $R^2_{CV}$ of the GAM is only 1 % higher for summer but 3 % lower for winter low flow when using the model without variable preselection. This suggests that non-linear processes, which are to be expected in such a heterogeneous study area as Austria, can be sufficiently captured by the superposition of linear terms, so there is no need to add non-linear effects or or to use a non-linear model. This is additionally supported by the performance of the three variable ranking methods. The non-linear approach of conditional forest leads only to a small reduction of the set of
predictors, with performance similar (winter $q95$) or worse (summer $q95$) than the two linear ranking methods.

## 6   Conclusions

In this study we investigated a broad range of statistical learning methods for a comprehensive dataset of 260 catchments in Austria. The results showed that all statistical learning models perform well and are therefore well suited for low flow region-alization. Performance is particularly high for summer low flow ($R^2_{CV} = 0.86$), but still leads to satisfactory results for winter
low flow ($R^2_{CV} = 0.70$). The best performing models are support vector machine regression (winter) and non-linear model based boosting (summer), but linear models exhibit similar prediction accuracy. No superior model could be found for both low flow processes, as relative differences between learning methods are actually small. The models perform similar or slightly better than a Top-kriging model that constitutes the current benchmark for the study area.

Variable preselection is shown on average to reduce the predictor set (on median) from 87 variables to 12 for winter and 8
for summer low flow. This is achieved by a small loss in performance, which is about 5 % for winter low flow, and only 1 % for summer low flow. The results suggest that variable preselection can help to create parsimonious learning models that





are easier to interpret and more robust when predicting at ungauged sites. The RF (summer) provides the most parsimonious model, which consists of only three variables and has a performance loss of less than 1 %.

Linear prediction models such as the linear model based boosting reveal high prediction accuracy. Non-linear terms were
shown to increase the goodness-of-fit of the models, but did not improve predictions at ungauged sites. Our results suggest that non-linear low flow relationships can be sufficiently captured by linear learning models, so there is no need to use more complex models or to add non-liner effects. This finding is confirmed by our variable ranking methods, where linear approaches seem to be sufficient for our estimation problem.

Variable rankings allow some conclusions about the importance of predictor variables. Topographic variables representing alti-
tude and slope are under the most highly ranked predictors for summer and winter low flows. Among meteorological predictors, characteristics representing snow-melt, aridity, and dry spells appear more important than precipitation characteristics. The best rated geological characteristics are the area fractions of limestone, Flysch and Quaternary sediments. Overall, topological, meteorological and catchment characteristics appear similar important for low flow regionalization. However, the interpretation of the variable ranking should be considered with caution as substituting top ranked variables in highly correlated data can lead
to similar performance.

Finally, the study shows that when performing low flow regionalization in a seasonal climate with a cold winter season, the temporal stratification into summer and winter low flows increases the predictive performance of all learning models. This suggests that conducting separate analyses of winter and summer low flows provides a data-efficient alternative to catchment grouping that is recommended otherwise.

*Code and data availability.* Data and code can be made available on personal request.





**Appendix A**

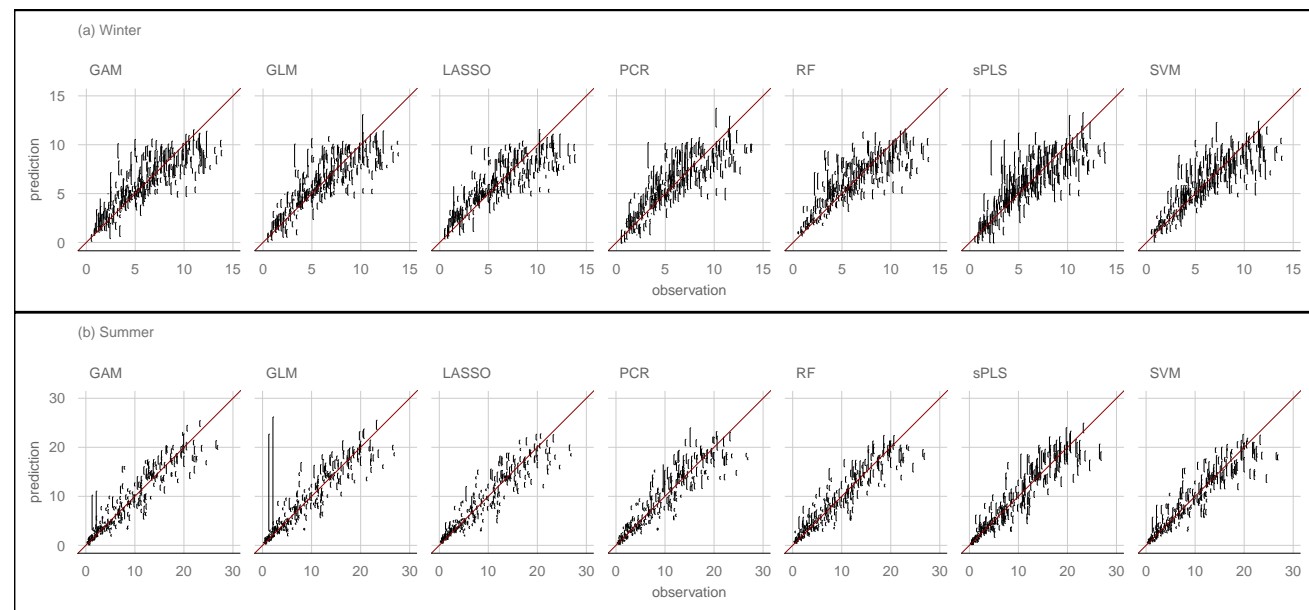

**Figure A1.** Error bars showing the range of predictions of the 10 CV-runs for each model without variable preselection. Two outliers are not shown for the summer PCR model, and the winter GLM, PCR, LASSO and GAM models to improve visual clarity.





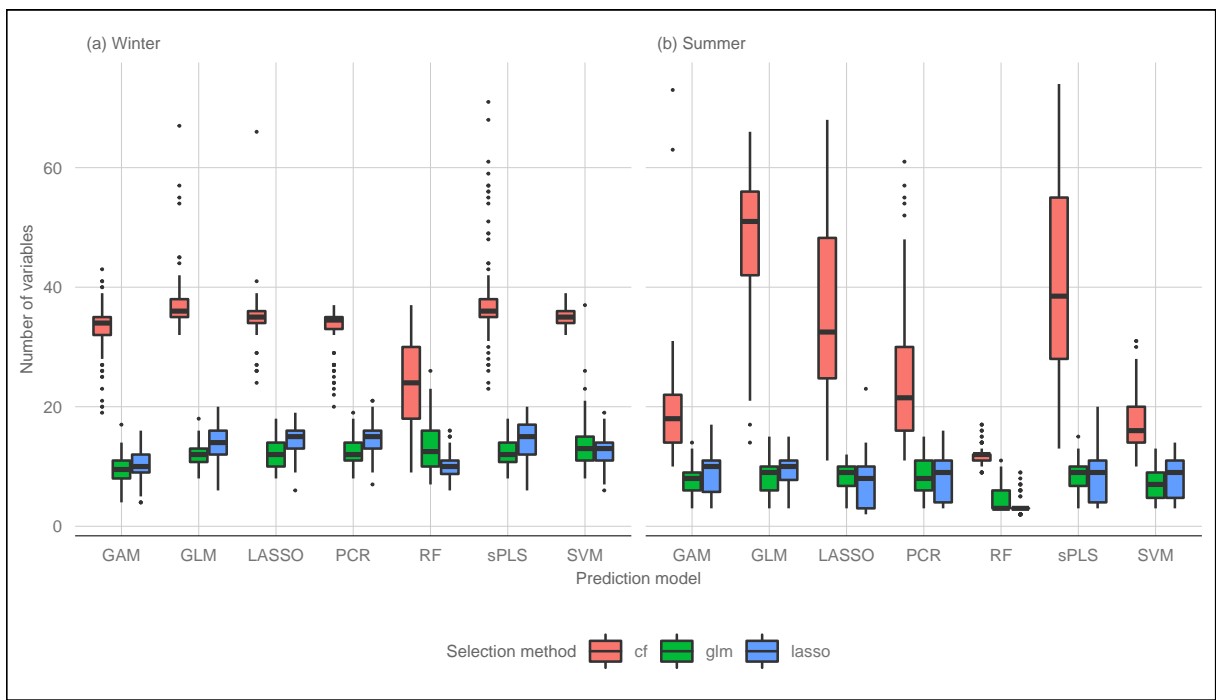

**Figure A2.** Boxplots of the number of variables selected in each CV-fold (100-times).







**Figure A3.** Count of all variable rankings of all 1000-iterations on a log-scale for summer low flow. The ten best variables are listed after their averaged rank.

*Author contributions.* JL designed the research layout and GL contributed to its conceptualization. JL performed the formal analyzes and prepared the draft paper. MM supported the analyzes. GL supervised the overall study. All authors contributed to the interpretation of the results and writing of the paper.

*Competing interests.* The authors declare that they have no conflict of interest.





*Acknowledgements.* J. Laimighofer is a recipient of a DOC fellowship (grant number 25819) of the Austrian Academy of Sciences, which is gratefully acknowledged for financial support. Data provision by the Central Institute for Meteorology and Geodynamics (ZAMG) and the Hydrographical Service of Austria (HZB) was highly appreciated. This research supports the work of the UNESCO-IHP VIII FRIEND-Water program.





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
