# Peer review of "Parsimonious statistical learning models for low flow estimation"

_Hydrology and Earth System Sciences, 2021_

## Author Response (AR1)

Reviewer #1

We want to thank Reviewer #1 for the valuable comments. Below you find our response and the changes we implemented in the document.

***Throughout the manuscript, the term parsimonious is used for characterizing the statistical learning models due to reducing the number of predictor variables to include the important ones. In statistical theory, the term parsimonious mainly refers to the number of parameters of the model, which is still large, e.g. in a random forest or boosting-based model with few predictor variables.***

We agree that the term parsimonious can be misleading if compared over all models, as the degree of freedom will be larger for simpler models as LASSO compared to a non-linear boosting approach. However, we argue that using less variables in a random forest model still reduces the complexity of the random forest model. So producing parsimonious models refers to reducing the number of variables for each individual model in our case. To avoid a misunderstanding we changed the term parsimonious to more parsimonious where it suits.

***It is self-contradicting to state that non-linear relationship can be captured by linear learning models (lines 14, 15). Actually, the term "non-linear relationship" cannot be defined. Furthermore, a more accurate classification of models considers their flexibility and interpretability, while the more flexible models can model better relationships, albeit this costs to their interpretability.***

The term „non-linear relationship" in this sentence was misleading, we meant non-linear hydrological processes, not non-linear coefficients in a linear model. We changed the sentence on line 13-15 to:

Line 13: A direct comparison of linear and non-linear models reveals that non-linear relationships can be sufficiently captured by linear learning models, so there is no need to use more complex models or to add non-liner effects.

Changed to: A direct comparison of linear and non-linear models reveals that non-linear processes can be sufficiently captured by linear learning models, so there is no need to use more complex models or to add non-linear effects.

***The classification "statistical" vs "machine" learning models is not clear either. For instance, Support Vector Regression or RF can be considered statistical learning models also.***

We did not want to make a distinction between machine learning models and statistical learning models as most models are indeed the same and the two terms are often used synonymously. We used statistical learning as we are more used to the terms used in statistical theory than to machine learning terms (e.g. coefficients vs. weights, variables vs. features).
To avoid a misunderstanding, we added the following clarification to line 24:

Line 24: Regression methods cover a wide spectrum of models and especially in the last decade there was gaining interest in statistical learning models in hydrology (Abrahart et al., 2012; Dawson and Wilby, 2001; Nearing et al., 2021; Solomatine and Ostfeld, 2008).

Changed to: Regression methods cover a wide spectrum of models and especially in the last decade there was gaining interest in statistical learning models in hydrology (Abrahart et al., 2012; Dawson and Wilby, 2001; Nearing et al., 2021; Solomatine and Ostfeld, 2008), with the terms statistical learning and machine learning being used synonymously.

Additionally we changed Support Vector Machine Regression to Support Vector Regression throughout the manuscript.

***Some claims related to the fact that studies mostly claim that non-linear models outperform linear ones could be relaxed. The literature includes studies claiming the opposite. Some of them can be found in the references list of the manuscript.***

Line 45: Tree based methods performed better in terms of point prediction for the CAMELS data set (Tyralis et al., 2021) or an Australian data set of 605 stations (Zhang et al., 2018).

Changed to: Tree based methods performed better in terms of point prediction for the CAMELS data set (Tyralis et al., 2021) or an Australian data set of 605 stations (Zhang et al., 2018), but both studies showed good performance for less complex linear models.

Line 51: A general tendency visible from most studies is, that more complex models seem to perform better than more parsimonious ones, making model interpretation difficult and plausibility of parameters hard to judge.

Changed to: Results of Worland et al. (2018) and Ferreira et al. (2021) indicate that more complex models seem to perform better than more parsimonious ones, making model interpretation difficult and plausibility of parameters hard to judge.

Reviewer #2

We want to thank Kolbjorn Engeland for his valuable comments. Below you find our response and the changes we implemented in the document.

***From the conclusions, it seems like one more research questions could be added: what are the most important predictor variables for low flows in Austria.***

We added the research question (iv), Which variables can be identified as the most important drivers of low flow for Austria? on line 88 of the manuscript.
One limitation that arises in answering this research question, is that we were working with a highly correlated dataset. Substituting the most important predictors will still lead to similar predictions (Efron 2020). In this context we tried to answer the question more on which groups of variables (e.g. geological, meteorological variables) tend to drive low flows in Austria.

***Units could be added to the x-axis.***

The unit [$l \, s^{-1} \, km^{-2}$] was added to the x-axis of Figure 2.

***Did you use 'residual catchments' or the total upstream catchment area for each gauging station when calculating low flows an catchment characteristics?***

Low flow and catchment characteristics were calculated on the total upstream catchment. To clarify this we added the following to Line 96:

Low flow and catchment characteristics are both based on the total upstream catchment of each gauging station.

***Why choose the grid point nearest the gaugng station. Would not catcment center of gravity be a better choice?***

In preliminary results, we found that this does not improve prediction accuracy (but also not impare). Furthermore, the data is currently used in a study for spatiotemporal low flow modeling, where we believe that the signal of extreme precipitation events is better estimated from a near grid point. So we have chosen the nearest grid point for a better consistency between our studies, but are aware that this may not be the best choice in terms of long term average low flow prediction.

***When using the 5x5 minutes gridded dataset, how did you account for altitude dependency of temperature (and possibly precipitation)?***

Temperature and precipitation was used from a 1x1 km gridded dataset, where we did not additionally account for any altitude dependency of the meteorological variable. Snowmelt is based on this daily datasets of temperature and precipitation and additionally uses the altitude, longitude and latitude for the calculation (Walter et al. 2005).
Snow fraction uses a 5x5 minutes gridded dataset. The dataset is already based on an interpolation procedure and a statistical relationship between temperature and solid precipitation (Chimani et al. 2011), therefore we did not further process the data. More accurate estimation of the fraction of snow and also of snowmelt could improve the model and is available in Austria by the SNOWGRID model (Olefs et al. 2014), but not for the full study period.

We added the following sentence on Line 117:

The dataset was not processed for a finer resolution, as it already relies on a statistical relationship of temperature to solid precipitation (Chimani et al. 2011).

***Line 140:***
***The GAM approach has previously been used for prediction of low flows in Ouarda et al (2018)***

As at this point we refer to the GAM only to the boosting approach, we changed the sentence to:

Line 140: This results in the (v) generalized additive boosting model (GAM), which has never been applied in a low flow study.

Changed to: This results in the (v) generalized additive boosting model (GAM). A maximum likelihood estimation of a generalized additive model in a regional frequency approach for low flow was already adapted by Ouarda et al. (2018).

***The use of the numbering of each step in sPLS is confusing.***

We removed the numbering in this section.

*Could be useful to repeat the complete name of RFE in a similar way that it is done for all the regression approaches.*

We added recursive feature elimination (RFE) on line 256.

*Figure 4:*
*It is difficult to see the colours of the green and red vertical lines. It might help if the colours are a bit brighter*

We updated Figure 4 by using thicker vertical lines.

*I would be happy if you add one sentence on how nested CV is used. Is it correct that all data are used on both loops of the CV, the only difference is that you make different splits for each of the loops?*

The nested CV – also referred to as double CV (Varmuza and Filzmoser 2009) – consists of two CV loops. Principally all datapoints are used as training set, validation set and test set, but not at the same time (Varmuza and Filzmoser 2009). In this first loop – we call it here now the outer loop - the dataset (260 stations) is split up into ten folds (k = 10), where each fold consists of 10% of the data (26 stations in our case). Nine of these 10 folds (90% of the data, 234 stations) are now used for model calibration, the $10^{th}$ fold is left out and only used for the final prediction. These 9 folds (of the outer loop) are now again splitted up 10 times into ten inner folds (j = 10, of the inner loop), which results in approximately 23 stations per fold of the inner loop. The full procedure of variable selection is now performed in the inner loop, where we determine the number of variables and the specific variables for the final prediction. This final prediction is made on the unseen outer fold. This nested CV is repeated (randomly) 10 times to get multiple prediction errors for each station.
One additional remark to the use of 10-folds in the inner loop. In some cases an inner loop of only 3 or 5 folds would be sufficient. We had to use 10 inner folds to reach convergence with the number of variables. Using 3 or 5 folds led to a too high variance for selecting the number of variables.

To clarify this nested CV scheme we added the following to the paragraph beginning line 290:

As one aim of this study is to get unbiased predictions (Solomatine and Ostfeld, 2008; Ambroise and McLachlan, 2002), we employ a nested CV scheme (referred to as double CV in Varmuza and Filzmoser 2009). The nested 10-fold CV consists of two loops, where the inner loop is used for model optimization and variable selection and the outer loop is used for independently evaluating the predictive performance of the so obtained models. In the outer loop the data is split into 10 folds, nine of which define the calibration data and the remaining one the test data. The calibration data are sent to the inner loop, where again, these data are split into ten folds, 9/10 of which being used for variable rankings and parameter estimation. The left out fold is used for estimating the RMSE for each candidate variable selection (l).

*Lines 435-440:*
*I think that also data quality might be an important factor when winter ow flows are calculated from data. My experience from Norwegian data is that water levels often increase due to freezing in the river. The standard rating curve cn therefore not be used to estimate discharge. In stead , discharge is estimated by gap-filling approaches like*

*interpolation,use of donor stations or simple hydrological or statistical models.*

This procedure is similar to the one used by the Austrian gauging authorities, but if measurement errors occurred through ice this is noticed in the meta data of the stations, and these stations were excluded from the analysis.

*In addition to the importance of different predictor variables, is it possible to summarize how they influence low flows? Do low flows increase or decrease with these predictors?*

This would be indeed a beneficial information, but not straightforward to implement. First of all, not all models used in our study offer the possibility of directly gain information about the positive or negative relationship between response and the individual predictor variable (e.g. SVR, RF).
Reducing the problem to the variable ranking methods that we applied, we could use the standardized coefficients of the three models. These coefficients change in the presence of other predictor variables, so the sign is dependent on the full model. Therefore, in our model setup it is at least not directly possible to assess the sign of the predictor variables, but we will take a look if a meaningful evaluation is possible.

We added the findings in the conclusion:

Variable rankings allow some conclusions about the importance of predictor variables. Topographic variables representing altitude and slope are under the most highly ranked predictors for summer and winter low flows. Specific low flow is mainly increasing with topographic predictors, except the percentage of slight slope in the catchment has a decreasing effect. Among meteorological predictors, characteristics representing snowmelt, aridity, and dry spells appear more important than precipitation characteristics. The aridity and number of dry days reduce specific low flow, whereas snowmelt has an increasing effect. The best rated geological characteristics are the area fractions of limestone, Flysch and Quaternary sediments. Limestone and Quaternary sediments, both lead to higher low flows, whereas Flysch has a decreasing effect. Overall, topological, meteorological and catchment characteristics appear similar important for low flow regionalization. However, the interpretation of the variable ranking should be considered with caution as substituting top ranked variables in highly correlated data can lead to similar performance.

**Additionally to the comments of both reviewers we have updated Table 1 for a better overview and agreement with previous studies of the variables and updated the figures accordingly.**

References:

Bradley Efron (2020) Prediction, Estimation, and Attribution, Journal of the American Statistical Association, 115:530, 636-655, DOI: 10.1080/01621459.2020.1762613.

M. Todd Walter, Erin S. Brooks, Donald K. McCool, Larry G. King, Myron Molnau, Jan Boll, Process-based snowmelt modeling: does it require more input data than temperature-index modeling?, Journal of Hydrology, Volume 300, Issues 1–4, 2005, https://doi.org/10.1016/j.jhydrol.2004.05.002.

Chimani, B., Böhm R., Matulla C., Ganekind M.: Development of a longterm dataset of solid/liquid precipitation Adv.Sci.Res,6,39-43, 2011http://www.adv-sci-res.net/6/39/2011/asr-6-39-2011.html

M. Olefs, W. Schöner, M. Suklitsch, C. Wittmann, B. Niedermoser, A. Neururer, A. Wurzer
SNOWGRID–A New Operational Snow Cover Model in Austria. International Snow Science
Workshop Proceedings 2013
(2014), pp. 38-45

Varmuza, K., & Filzmoser, P. (2009). Introduction to Multivariate Statistical Analysis in Chemometrics
(1st ed.). CRC Press. https://doi.org/10.1201/9781420059496